# Health Literacy Gaps Across Language Groups: A Population-Based Assessment in Alto Adige/South Tyrol, Italy

**DOI:** 10.3390/ejihpe15080153

**Published:** 2025-08-09

**Authors:** Dietmar Ausserhofer, Verena Barbieri, Stefano Lombardo, Timon Gärtner, Klaus Eisendle, Giuliano Piccoliori, Adolf Engl, Christian J. Wiedermann

**Affiliations:** 1Institute of General Practice and Public Health, Claudiana—College of Health Professions, 39100 Bolzano, Italy; 2Claudiana Research, Claudiana—College of Health Professions, 39100 Bolzano, Italy; 3Provincial Institute for Statistics of the Autonomous Province of Bolzano—Alto Adige/South Tyrol (ASTAT), 39100 Bolzano, Italy; 4Directorate, Claudiana—College of Health Professions, 39100 Bolzano, Italy; klaus.eisendle@claudiana.bz.it

**Keywords:** health literacy, multilingual regions, health promotion, disease prevention, health equity

## Abstract

Health literacy is crucial for effectively navigating health systems and promoting equitable health outcomes. Multilingual and culturally dual regions present unique challenges for health communication; however, disparities in health literacy within such contexts remain insufficiently explored. This study constitutes the first population-based assessment of health literacy in Alto Adige/South Tyrol, a bilingual province in northern Italy, utilizing the validated HLS-EU-Q16 instrument. A stratified random sample of 2090 residents aged 18 and older was surveyed in 2024. Weighted analyses ensured population representativeness, and scores were analyzed overall, by domain (health care, disease prevention, health promotion), and by language group (German, Italian, multilingual). Regression models incorporating sociodemographic and health-related covariates were employed to identify predictors of health literacy. Half of the population (50.0%) exhibited problematic or inadequate health literacy, with significant differences observed across language groups. Italian speakers demonstrated the highest scores, whereas German speakers scored lowest overall. These differences remained significant after adjustment for age, education, chronic illness, and professional background. Domain-specific analyses revealed distinct patterns: German-speaking respondents scored particularly low in the health promotion domain, while multilingual individuals achieved the highest scores in the prevention and promotion domains. Education level and language background emerged as the strongest predictors of health literacy, while most other covariates exhibited limited explanatory power. The findings underscore the necessity for language-sensitive and domain-specific interventions, highlighting health literacy as both a personal skill and a structural responsibility.

## 1. Introduction

Health literacy (HL) refers to an individual’s ability to access, understand, evaluate, and apply health information for healthcare decisions, disease prevention, and health promotion ([33]). HL encompasses practical abilities such as literacy and numeracy, along with the mental and social skills essential for navigating healthcare systems and managing one’s own health ([20]).

The European Health Literacy Survey (HLS-EU), launched in 2009, developed a comprehensive measurement framework ([33]; [14]). The HLS-EU conceptual model frames HL across healthcare, disease prevention, and health promotion domains along four stages: accessing, understanding, appraising, and applying health information ([24]). This led to the validated HLS-EU-Q47 questionnaire, which measures general HL across European countries. A shorter version, the HLS-EU-Q16, was later validated to facilitate large-scale research. This 16-item scale maintains the conceptual framework while reducing the respondent burden, making it suitable for epidemiological research across diverse settings ([34]).

HL is now recognized as a key social determinant of health ([5]). Evidence links low HL to poorer health outcomes, reduced preventive service use, and higher healthcare costs. Higher HL correlates with better health outcomes, patient empowerment, and navigation of the health system. Addressing HL disparities is crucial for promoting health equity in complex healthcare environments ([14]; [5]). While HL measurement tools such as the HLS-EU-Q16 have facilitated international comparisons, interpreting national differences remains complex. Prior research has raised questions about whether comparatively low HL scores observed in Italy may reflect cultural differences, system complexity, or response tendencies related to linguistic phrasing ([16]; [23]). However, these hypotheses have not been directly tested, and such methodological investigations were beyond the scope of the present study.

Health literacy levels vary across Europe, as shown in the European Health Literacy Survey (HLS-EU 2012) ([33]) and the WHO Health Literacy Survey 2019 (HLS19) ([25]). These studies revealed higher literacy levels in northern and western Europe and lower levels in southern and eastern European countries. The original HLS-EU 2012 study found limited health literacy ranging from 29% in the Netherlands to 62% in Bulgaria ([33]). The more recent WHO Health Literacy Survey 2019 (HLS19), coordinated by the M-POHL network, confirmed these patterns and showed that limited health literacy remained more prevalent among older adults, those with lower education, and migrants ([25]; [14]).

Germany participated in both survey waves but also conducted a separate national study, HLS-GER 1, in 2014. This study applied the HLS-EU instruments independently and reported that 42% of adults had limited health literacy, with only 7.3% reaching “excellent” levels—figures that were below the EU average at the time ([28]). In Austria, national HL levels improved between 2011 and 2020, potentially as a result of national HL initiatives ([12]).

Italy continues to rank below the European average, with 57.7% of adults reporting limited health literacy according to HLS19. These national disparities highlight the need to investigate HL variation at regional levels, especially in multilingual contexts such as Alto Adige/South Tyrol. Disparities were notable among older individuals, low-income groups, and those with poor health ([25]; [23]). However, recent validation work on the HLS-EU-Q16 and Q6 in the Italian context ([16]) suggests that language and cultural context may influence how HL is reported. The Italian version of the HLS-EU tools has shown good reliability but tends to yield lower scores compared to German-language versions, possibly due to cultural and cognitive response patterns or the phrasing of certain items. These instrument-related differences underscore the importance of interpreting national HL estimates, such as those from Italy, in light of linguistic adaptation and cultural framing. While our study did not directly assess the linguistic appropriateness of health information, it aimed to explore health literacy patterns in a multilingual context, using the HLS-EU-Q16 in its validated Italian and German versions as applied in Alto Adige/South Tyrol.

This study focuses on Alto Adige/South Tyrol (Autonomous Province of Bolzano), a multilingual region in northern Italy where both German and Italian are official languages ([27]). The region offers a unique context for HL research: two large language groups coexist under a shared health, education, and media infrastructure, allowing for analysis of language-related HL differences while holding institutional factors constant. The region’s location and history have created a distinct sociocultural profile that combines central European and Mediterranean features. Alto Adige/South Tyrol differs from Italy in terms of education, economic indicators, and healthcare organization, operating an autonomous, bilingual healthcare system that provides services in German and Italian ([27]). The region’s diverse media landscape and cultural norms around health may influence HL outcomes differently from national-level data.

While studies have identified education, age, and migration background as major determinants of limited HL ([21]), little is known about these patterns in regions with parallel language communities. Although HL has gained attention in public health research, gaps remain in multilingual regions. Most HL studies in Italy use nationally aggregated data ([23]), which do not capture regional heterogeneity. No international publication has systematically examined health literacy in Alto Adige/South Tyrol using a European framework to date. Despite the region’s unique linguistic composition, empirical evidence on HL differences within Alto Adige/South Tyrol remains lacking. It remains unclear how language group affiliation influences HL in Alto Adige/South Tyrol’s bilingual setting.

This population-based study aimed to describe the distribution of general HL in a representative adult sample in Alto Adige/South Tyrol using the HLS-EU-Q16 instrument. Specifically, it examined differences in HL levels between the German- and Italian-speaking language groups and explored sociodemographic and health-related predictors of HL. While prior reports have analyzed related outcomes such as vaccine uptake ([2]) and health information use and trust ([42]), the present analysis offers the first in-depth profile of HL across population subgroups in this multilingual region.

## 2. Methods

### 2.1. Study Design and Participants

This study analyzed data from a cross-sectional, population-based health survey conducted between March and May 2024 in the Autonomous Province of Bolzano (Alto Adige/South Tyrol), Italy. The province is known for its multilingual population (German, Italian, and Ladin) and decentralized healthcare system. The Provincial Institute of Statistics (ASTAT) designed the survey in collaboration with the Institute of General Practice and Public Health.

A random sample of 4000 residents aged 18 years or older was selected from the provincial population registries. Stratification ensured representation across age groups (18–34, 35–54, and 55+), sex, citizenship status, and municipality of residence. Invitations to participate were sent by post and included a choice between paper-based and online response. All materials were provided in German and Italian. Participation was anonymous and voluntary, and informed consent was obtained from all respondents.

A total of 2090 individuals completed the survey, yielding a 52.3% response rate. Post-stratification weights were computed using the ReGenesees software (version 2.3, ISTAT, Rome, Italy) by the ASTAT to align the sample with the demographic structure of the adult population in Alto Adige/South Tyrol. Weighting accounted for age group, sex, citizenship status, and municipality of residence. The final weight variable (Weights) integrated two calibration components and was normalized for analytic use. Its values ranged from 0.70 to 3.04, with a mean of 1.13 and a standard deviation of 0.52. All reported descriptive and regression analyses were weighted accordingly. Due to population weighting, some tables report weighted effective participant numbers that differ from the actual respondent count.

### 2.2. Measures

#### 2.2.1. Sociodemographic and Health Characteristics

Participants reported their sex (male/female), age (in years), and primary language group (German, Italian, Ladin, other, or more than one). For some analytical purposes, the Ladin and other language groups were combined into a single category. Educational attainment was assessed using four levels: (1) middle school, (2) vocational/technical training, (3) high school diploma, and (4) university degrees.

Household composition was derived from responses regarding cohabitation with a partner, children, parents, or other relatives. Based on these items, a binary variable was constructed to indicate whether the respondent lived alone. Chronic illness status was self-reported using a checklist of prevalent conditions, including cardiovascular, metabolic, respiratory, and mental health disorders. A binary variable was created to reflect the presence or absence of any of the listed conditions. Additional variables included employment in the health or social care sector (yes/no) and citizenship (Italian vs. non-Italian).

#### 2.2.2. Health Literacy

General health literacy was assessed using a 16-item version of the HLS-EU-Q16, which is available in both German ([34]) and Italian ([17], [16]) versions. While the development of the HLS-EU-Q47 and its short form, the HLS-EU-Q16, was grounded in an extensive conceptual framework, multiple studies have confirmed their psychometric validity. In particular, the Italian version of the HLS-EU-Q16 has demonstrated good internal consistency and acceptable construct validity and has been used in several large-scale population studies ([17], [16]). These findings support its use as a valid instrument for assessing general health literacy in the Italian context. This short scale captures the ability to access, understand, appraise, and apply health information across the domains of healthcare, disease prevention, and health promotion. Items were scored according to standard guidelines, and respondents who answered fewer than 13 items were excluded from the HL scoring. For respondents who answered 13 to 16 items, a general HL index (range 0–16) was computed and categorized as follows:0–8: inadequate;9–12: problematic;13–16: sufficient.

Respondents with incomplete HL data were identified as a separate “missing” category for descriptive purposes but were excluded from the correlation and regression analyses.

To explore domain-specific patterns of HL, three theoretically grounded subscales from the HLS-EU-Q16 were constructed, corresponding to the original model’s conceptual domains: health care (HC), disease prevention (DP), and health promotion (HP) ([33]). Although the HLS-EU-Q16 was designed as a short-form index to yield a global HL score, previous research has demonstrated the feasibility of post hoc assignment of items to the three domains based on their semantic and conceptual content, as derived from the full HLS-EU-Q47 scale ([24]).

Each of the 16 dichotomized items (0 = “very difficult/difficult,” 1 = “easy/very easy”) was assigned to one of the three domains as follows (Appendix A):Health care (HC): items Q2, Q3, Q4, Q5, Q8;Disease prevention (DP): items Q1, Q6, Q7, Q10;Health promotion (HP): items Q9, Q11, Q12, Q13, Q14, Q15, Q16.

Domain-specific sum scores were computed only for respondents with valid overall HLS-EU-Q16 scores (i.e., ≥13 valid items). For each domain, the score was calculated by summing all available dichotomized items (0 = “very difficult/difficult”, 1 = “easy/very easy”) assigned to that domain. If one or more domain items were missing, the score was based on the sum of the remaining valid responses. This approach, consistent with previous studies ([17]; [24]), enables inclusion of partially completed domains while maintaining the threshold of ≥13 items at the total scale level.

Internal consistency of the domain-specific subscales derived from the HLS-EU-Q16 was assessed using Cronbach’s alpha. Following common guidelines, alpha values above 0.70 are considered acceptable, values between 0.60 and 0.70 indicate moderate reliability, and values below 0.60 reflect limited internal consistency ([36]). However, Cronbach’s alpha is sensitive to the number of items in a scale and may underestimate reliability in short subscales. Therefore, alpha values in this study were interpreted with caution, particularly for domains with fewer items.

### 2.3. Statistical Analysis

All analyses were conducted using IBM SPSS Statistics, Version 29.0.2.0 (20). Statistical significance was set at *p* < 0.05 (two-tailed).

Descriptive statistics were used to characterize the sample according to the language group. Categorical variables were summarized as weighted proportions and frequencies, and differences between groups were tested using the chi-square (χ^2^) test. Continuous variables were summarized as weighted means and standard deviations (SDs) and compared using one-way analysis of variance (ANOVA). All descriptive estimates were weighted using post-stratification weights to reflect the demographic structure of the adult population of Alto Adige/South Tyrol. Effect sizes are reported as η^2^ (eta-squared) for ANOVA and Cramér’s V for chi-squared tests (η^2^: small ≥ 0.01, medium ≥ 0.06, large ≥ 0.14; Cramér’s V: small ≥ 0.10, medium ≥ 0.30, large ≥ 0.50).

Weighted Spearman rank correlations were calculated to assess bivariate associations between HL and sociodemographic or health-related factors. Correlation coefficients (ρ) were accompanied by *p*-values and Fisher z-transformed effect sizes to allow for the interpretation of magnitude ([8]).

To examine the predictors of HL, a weighted multiple linear regression using the HLS-EU-Q16 index as the continuous outcome variable was performed. The model included age, sex, education, living situation, chronic illness, employment in the health or social care sector, citizenship status, and language group. Population weights were applied to reflect the demographic structures of Alto Adige/South Tyrol. Unstandardized coefficients (B) reflect changes in the health literacy score per unit increase in the predictor. Standardized coefficients (β) allow for comparing the strength of associations across predictors measured on different scales. Variance inflation factors (VIFs) were examined. The model was evaluated using R^2^, F-statistics, and 95% confidence intervals.

## 3. Results

### 3.1. Sociodemographic and Health Characteristics by Language Group

Table 1 presents the weighted sociodemographic and health-related characteristics of the sample stratified by language group. The sample included 2090 participants, with 63.7% assigned to the German-speaking group, 22.9% to the Italian-speaking group, and 13.4% to the “Other languages” group.

The sex distribution did not differ significantly between the groups. Women comprised approximately half of each subgroup in the study. Age differed significantly between the language groups (η^2^ = 0.030), corresponding to a small effect size. The effect size for the age group comparison (Cramér’s V = 0.137) indicated a moderate association based on the conventional thresholds. The Italian-speaking group was the oldest on average (mean = 57.6 years), while the “Other languages” group was the youngest (mean = 45.8 years).

The education level varied significantly, representing a small-to-moderate effect. Vocational education was most common in the German-speaking group (37.6%), while university education was most prevalent in the “Other languages” group (28.2%).

The most pronounced difference across groups was citizenship status, which represented a large effect size. Over half (50.7%) of the “Other” group reported non-Italian citizenship compared to only 0.6% of the Italian group and 4.5% of the German group.

No significant differences were found in the proportion of participants working in the health or social sectors. Living alone was significantly more frequent in the Italian-speaking group (22.8%) and least frequent in the “Other” group (14.2%) (Cramér’s V = 0.075, small effect). Chronic illness was most reported in the Italian group (43.8%) and least in the “Other” group (26.6%), with a small to moderate effect. Health status also differed across groups (η^2^ = 0.009), although the effect was negligible based on self-rated health on a 0–100 scale.

The mean and SD of the overall HLS-EU-Q16 score was approximately 11.9 ± 3.4 (*n* = 1651). A further 21.0% of respondents did not provide sufficient responses to calculate the HLS-EU-Q16 score. Health literacy scores (HLS-EU-Q16 index) differed significantly (η^2^ = 0.018), indicating a small effect size. On average, the German-speaking group scored the lowest (mean = 11.6), while the Italian-speaking and “Other” groups showed higher scores (12.5 and 12.4, respectively). In categorical levels, the German-speaking group had the highest proportion of inadequate health literacy (18.7%) and the lowest proportion of sufficient health literacy (43.9%), whereas the Italian-speaking group showed the highest proportion of sufficient health literacy (60.5%). The effect size for health literacy levels (Cramér’s V = 0.105) indicated a small-to-moderate association. Compared to the other language groups, the Italian-speaking population was older on average and had higher proportions of individuals living alone and reporting chronic illness.

The “Other” group was characterized by younger age, higher education, and a much higher rate of non-Italian citizenship than the German or Italian groups. This group exhibited comparable or better health literacy scores than those of Italian and German speakers. A more detailed examination of this group, subdivided into Ladin speakers, speakers of other languages, and individuals who reported speaking more than one language, revealed internal differences (Appendix A). Among participants who reported a primary language other than German or Italian (*n* = 280), no significant differences in overall health literacy scores were observed across the three subgroups: Ladin, other, and more than one language. The mean HLS-EU-Q16 index was similar across groups (Ladin: 12.3 ± 2.6, *n* = 79; other: 12.6 ± 3.3, *n* = 145; more than one: 12.8 ± 2.8, *n* = 56), with a non-significant Kruskal–Wallis test result (*p* = 0.381) and a negligible effect size (η^2^ = 0.001).

When examining categorical HL levels, significant differences emerged across the three subgroups (χ^2^ = 27.98, df = 6, *p* < 0.001), with a moderate effect size (Cramér’s V = 0.246). The Ladin-speaking group showed the highest problematic HL (37.2%) and lower sufficient HL (42.3%), while the “Other” subgroup had the lowest limited HL (20.0%) and highest missing values (35.2%). The “More than one” group showed balanced proportions, with half having sufficient HL and 12.5% missing values. These differences were partly influenced by varying valid HL responses, with the Ladin subgroup having the fewest missing cases (11.5%) (Appendix A).

Post hoc comparisons of HL level distributions showed significant differences between Ladin and other language groups (χ^2^ = 15.62, df = 2, *p* < 0.001), with moderate effect size (Cramér’s V = 0.355). The Ladin group had higher proportions of problematic HL (37.2% vs. 12.4%) and lower sufficient HL (42.3% vs. 44.8%), though the difference in sufficient HL was small. The “Other” group showed higher missing HL scores (35.2% vs. 11.5%). A significant difference existed between Ladin and “More than one language” groups (χ^2^ = 7.56, df = 2, *p* = 0.023), with small-to-moderate effect size (Cramér’s V = 0.265). The “More than one” group had fewer respondents with inadequate HL (3.6%) and more classified as sufficient (50.0%) compared to 9.0% and 42.3%, respectively, in the Ladin group. No significant differences were found between “Other” and “More than one” groups (χ^2^ = 1.78, df = 2, *p* = 0.410; Cramér’s V = 0.136), suggesting similar HL distributions.

### 3.2. Bivariate Associations Between Health Literacy Index and Sociodemographic Factors

Table 2 presents the weighted Spearman correlation coefficients between the continuous HLS-EU-Q16 index and key sociodemographic and health-related variables. The strongest positive association was observed for the Italian language group compared to the German group (ρ = 0.143, *p* < 0.001), corresponding to a small effect size. Weaker but statistically significant associations with health literacy were also found for higher education level (ρ = 0.069, *p* = 0.004), working in the health or social care sector (ρ = 0.058, *p* = 0.015), living alone (ρ = 0.049, *p* = 0.043), and having a chronic illness (ρ = −0.056, *p* = 0.020). These correlations were all of negligible magnitude.

Gender, citizenship status, and age showed no statistically significant associations with health literacy, and the corresponding correlation coefficients were close to zero. Overall, these findings suggest that language group membership, educational attainment, and work context are more meaningfully associated with health literacy than basic demographic characteristics alone.

Intercorrelations between predictor variables are shown in Appendix A. Age was strongly and negatively correlated with education level (ρ = −0.392, *p* < 0.01) and positively correlated with the presence of chronic illness (ρ = 0.403, *p* < 0.01), both indicating large effect sizes. Smaller, but statistically significant associations, were found between education and working in the health or social care sector (ρ = 0.192) and between gender and employment in this sector (ρ = 0.165), both with small effect sizes. Most other associations between sociodemographic determinants, including language group, citizenship, and living alone, were statistically significant but reflected negligible to small effect sizes. These findings suggest that while some predictors of HL are interrelated, multicollinearity is unlikely to bias the main correlation results.

### 3.3. Predictors of Health Literacy

A multiple linear regression was conducted to examine the associations between HL and a set of sociodemographic and health-related predictors in a population-weighted sample (N = 1436) (Table 3). The overall model was statistically significant (F(8, 1427) = 5.813, *p* < 0.001) and explained 3.2% of the variance in HL scores (adjusted R^2^ = 0.026).

Among the predictors, speaking Italian as the primary language (B = 0.954, β = 0.124, *p* < 0.001) was significantly associated with higher HL, as was higher educational attainment (B = 0.204, β = 0.061, *p* = 0.036). Having a chronic disease was negatively associated with health literacy (B = −0.506, β = −0.070, *p* = 0.015). Age showed a marginally significant positive association (B = 0.012, β = 0.059, *p* = 0.057). All other predictors, including gender, citizenship, living alone, and working in the health or social sector, were not significantly associated with health literacy in the adjusted model (all *p* > 0.05).

Collinearity diagnostics indicated acceptable tolerance and variance inflation factors (VIFs all <1.40), suggesting no problematic multicollinearity among the independent variables.

### 3.4. Health Literacy Domains by Language Group

Internal consistency of the domain-specific health literacy questions was assessed using Cronbach’s alpha for each of the three post hoc subscales derived from the HLS-EU-Q16. The health promotion domain (seven items) demonstrated the highest internal consistency (α = 0.691), followed by the health care domain (5 items; α = 0.665). The disease prevention domain (4 items) showed lower internal consistency (α = 0.566), which falls below commonly accepted thresholds and indicates limited internal consistency. These results should be interpreted with caution, particularly for domains with fewer items, and are reported here to support exploratory interpretation.

These values were calculated using unweighted data, consistent with prior applications of the HLS-EU-Q16 in population-based surveys. However, the limited number of items per domain and the fact that the internal consistency was only moderate, particularly in the disease prevention domain, suggest that these subscales should be interpreted with caution and are not suited for diagnostic purposes or detailed domain-level comparisons.

Among the 1649 respondents with a valid HL index score, significant differences emerged across language groups in all three HL domains: health care, disease prevention, and health promotion (Table 4). Chi-square tests indicated that score distributions varied significantly between groups in each domain (all *p* < 0.05), with small effect sizes measured by Cramér’s V (range: 0.081–0.086).

Italian-speaking respondents consistently demonstrated higher domain-specific HL, particularly in the health care and disease prevention domains. Multilingual individuals also scored highly, especially in health promotion, where over one-third achieved the top score. Ladin speakers showed a distinct pattern, aligning closely with Italian and multilingual respondents in disease prevention but trailing in health promotion.

## 4. Discussion

This population-based study is the first to provide a comparative assessment of HL in Alto Adige/South Tyrol using the validated HLS-EU-Q16 questionnaire. The findings reveal significant differences in HL levels between language groups within this multilingual region, with the German-speaking majority exhibiting lower average scores and a higher prevalence of limited HL than Italian speakers. These disparities persisted even after adjusting for sociodemographic and health-related variables of interest. Although Italian-language group membership emerged as the strongest independent predictor of higher HL, the explanatory power of the overall model was modest, suggesting that language affiliation alone does not fully account for the observed differences. Other established determinants, such as education, chronic illness, and professional background in health or social care, showed only weak or borderline associations. The absence of a clear explanatory pattern in the correlation and regression analyses highlights the complexity of HL disparities in culturally and institutionally dual regions. This suggests that unmeasured contextual factors, such as the accessibility and linguistic appropriateness of health information, regional media use, or sociocultural norms around health, may play an important role in shaping population HL profiles in Alto Adige/South Tyrol.

### 4.1. Comparison with the Previous Literature

The results are broadly consistent with earlier HL research from Italy and its neighboring countries. In the HLS19 survey, Italy showed a national prevalence of limited HL (problematic or inadequate) of 57.7% ([6]), higher than Alto Adige/South Tyrol’s 50.0% observed in this study. This correlates with regional educational and socioeconomic disparities, with Alto Adige/South Tyrol often outperforming southern Italian regions in terms of health and education indicators ([26]; [38]; [9]). These differences reflect broader patterns of regional educational disparities. Research has shown that relative poverty is a more significant predictor of educational success across Italian regions than gross domestic product or background alone ([10]). This finding highlights that areas such as Alto Adige/South Tyrol can still experience internal disparities due to educational tracking and structural opportunity gaps. Similarly, the findings suggest that vocational education may be less aligned with the cognitive and evaluative demands of health literacy as captured by the HLS-EU-Q16 instrument. In contrast, academic education may better support the development of critical competencies for navigating health information.

In Slovenia, a neighboring state to northern Italy, a limited HL of 48% (HLS-SI19) was observed ([39]). The Swiss Federal Office of Public Health conducted a national health literacy survey on citizens over the age of 15; the results of the 2020 study revealed that approximately 38% of the population had limited HL ([11]); in Germany, 42% of adults had limited HL (HLS-GER1) ([29]), and in Austria, 44% of adults had limited HL (HLS19-AT) ([30]). Contrary to the findings from predominantly German-speaking populations with moderate HL levels, the German-speaking subgroup in Alto Adige/South Tyrol reported the lowest health literacy scores, with 50% classified as having limited HL. This unexpected pattern may reflect structural or cultural factors specific to Alto Adige/South Tyrol, such as divergent health information channels, media usage, or patient expectations.

Previous studies have suggested that German speakers in Alto Adige/South Tyrol rely more on traditional or interpersonal sources of health information, whereas Italian speakers more frequently engage with digital platforms ([42]; [1]). Age and education patterns also followed established trends ([22]). Younger and better-educated respondents in Alto Adige/South Tyrol tended to show higher HL, although the effect sizes were small.

The impact of migration background, approximated by a combination of non-Italian citizenship and the use of a language other than German or Italian, was more complex. While European studies often report lower HL among migrant populations ([41]; [19]), data here showed only a marginal, nonsignificant positive association. This deviation may reflect the heterogeneity of the “Other” group, which includes not only participants with a potential migration background but also Ladin speakers, a recognized linguistic minority comprising approximately 4.5% of Alto Adige/South Tyrol’s population ([15]), and individuals who report speaking multiple languages. Members of this group also had the highest proportion of university education, which may help explain their relatively high HL score. Their educational and linguistic profiles suggest that they are not directly comparable to nationally aggregated samples of migrants, who often face compounded barriers in accessing health information. In the Alto Adige/South Tyrolean context, multilingualism, institutional recognition of Ladin as a protected language, and access to bilingual or culturally adapted services may mitigate the typical disadvantages associated with minorities and migration backgrounds in HL research. This highlights the need to consider local sociolinguistic dynamics when interpreting HL outcomes in diverse populations in the future.

### 4.2. Interpretation of Language Group Differences

The observed HL disparities between German- and Italian-speaking residents are not easily explained by individual-level factors. Italian-speaking respondents were older, more likely to live alone, and more frequently reported chronic illness, factors typically associated with lower HL. Yet this group showed significantly higher HL scores. This suggests that additional factors may be involved. Prior studies using the same population sample have reported greater digital media use and higher trust in health professionals among Italian speakers, which may partly account for these differences ([1]; [42]). Familiarity with the national health system’s terminology and communication style may also play a role, but this remains to be directly tested in future research.

While education is a known determinant of health literacy, findings suggest that the type of educational attainment is important in determining HL. The German-speaking group had the highest share of vocational school graduates (37.6%), whereas the Italian-speaking group showed higher representation in academic pathways, particularly university education (24.4% vs. 18.2%). This may partly explain the higher HL scores observed in Italian speakers.

Interestingly, the group reporting the use of more than one language at home displayed relatively high levels of HL. Although this group is heterogeneous, it is conceivable that individual multilingualism or multicultural exposure may enhance informational flexibility and health-related navigation skills. While not directly explored in the present study, this observation suggests a promising direction for future research into the potential protective role of individual multiculturalism for health literacy.

### 4.3. Domain-Specific Disparities in International Context

Beyond the overall HL index, an analysis using the HLS-EU-Q16 highlighted notable disparities among language groups in perceived skills within the domains of health care (HC), disease prevention (DP), and health promotion (HP). Of the 1649 respondents with valid HL scores, those who spoke Italian consistently achieved higher maximum scores, especially in health care (40.2%) and disease prevention (62.4%). Multilingual participants performed equally well or better in disease prevention (69.6%) and health promotion (37.0%), indicating that access to diverse health information sources may improve engagement and understanding. Ladin speakers matched the Italian group in disease prevention (61.6%) but had lower scores in health promotion (21.9%). German-speaking respondents consistently had the lowest top scores across all three domains, particularly in health promotion (23.7%), highlighting a possible deficiency in navigating and applying health-related information outside clinical settings.

Domain-level differences highlight that health literacy varies across areas; individuals may excel in disease prevention but struggle with health promotion. In multilingual regions such as Alto Adige/South Tyrol, public health strategies must address these profiles through tailored interventions that consider both language accessibility and specific domain needs, particularly in health promotion content for German speakers. The study suggests that cultural, linguistic, and system-level factors (e.g., information availability, plain language use, and media targeting) may play a more decisive role in HL than previously thought.

The domain-specific HL patterns observed in this study reflect broader findings in Europe and other international settings. Previous research using the HLS-EU-Q16 and HLS-EU-Q47 has demonstrated that certain domains, particularly health promotion, tend to yield lower scores among vulnerable subgroups such as migrants, individuals with limited education, and those with low social integration ([33]; [18]; [4]). While a social gradient is consistently observed across all domains, disparities appear to be more pronounced in health promotion and disease prevention than in health care, where formal service contact may offer more structured support ([3]).

Domain interrelation has also been documented; individuals with limited skills in one domain frequently report challenges across others, indicating that vulnerabilities in HL are multifaceted rather than isolated ([3]; [37]). This supports our finding that German-speaking participants, despite representing the regional majority, exhibited lower top-level HL scores across all domains, suggesting systemic and cultural barriers beyond individual characteristics. Furthermore, the relatively strong performance of multilingual respondents in the disease prevention and health promotion domains aligns with studies highlighting the potential cognitive and informational advantages of navigating multiple linguistic environments ([4]).

Taken together, these international comparisons support the relevance of benchmarking regional HL patterns against broader national and European data, one of the aims of this study. They also suggest that future HL monitoring in Alto Adige/South Tyrol may benefit from more domain-sensitive assessments and attention to contextual factors such as information access, cultural alignment of messaging, and trust in information sources, which were not directly measured here but have been shown to influence HL outcomes in related work ([42]).

### 4.4. Implications for Public Health

The results of this study have significant and actionable implications for public health policy in multilingual and institutionally dual regions, such as Alto Adige/South Tyrol. The pronounced disparities in both overall and domain-specific HL among language groups, despite comparable demographic and educational characteristics, indicate that universal interventions may not be equally effective across all population segments. Therefore, it is important to strengthen language-sensitive and culturally responsive approaches to ensure equitable access to health information and services.

In Alto Adige/South Tyrol, public institutions adhere to the principle of official bilingualism, and communication strategies are generally designed to treat both German and Italian speakers equally. Translations are coordinated centrally to prevent any perception of linguistic favoritism, and language-specific differentiation is typically avoided in public messaging. However, the consistently lower HL scores among German-speaking residents, particularly in the domains of health promotion and disease prevention, suggest that formally equal communication may not fully achieve functional equity across groups. In line with prior recommendations for multilingual health systems ([32]; [40]), potential refinements could include the systematic use of plain language in German-language materials, targeted health promotion via trusted German-language media, and HL-sensitive training for healthcare professionals. These suggestions aim to support all groups more effectively, while respecting the region’s commitment to institutional language balance.

Additionally, educational pathways that prioritize applied knowledge, such as vocational schools, should be enhanced with curriculum components that bolster critical, health-related decision-making and system navigation skills. These approaches are supported by international evidence showing that tailored, culturally adapted, and community-engaged health literacy interventions are more effective than generic strategies, particularly among disadvantaged or linguistically diverse populations ([31]; [35]). Incorporating such elements into regional policy and education systems may be key to closing the persistent HL gaps in Alto Adige/South Tyrol. Multi-component, person-centered strategies, especially those co-designed with target communities, consistently improve HL outcomes among vulnerable subgroups ([13]; [7]). These insights argue for embedding tailored HL efforts into Alto Adige/South Tyrol’s institutional and educational structures to maximize their relevance and effectiveness.

The strong HL performance of multilingual and Ladin-speaking respondents suggests that structural investments in bilingualism, minority language inclusion, and culturally adapted services can serve as effective levers for improving HL. Therefore, health policy in Alto Adige/South Tyrol should continue to support and expand initiatives that provide multilingual and locally relevant health information. These include integrated communication platforms, co-designed materials with community input, and digital health tools that reflect the region’s linguistic diversity and cultural nuances.

At a broader level, the findings underscore that HL must be addressed not only as an individual competency but also as a structural characteristic of the health system. Health literacy responsiveness should be integrated into the design of all health-related services and communication strategies, guided by frameworks such as the WHO’s health-literate organization model ([25]). Routine HL monitoring, disaggregated by language, education, and domain, should be institutionalized to track health disparities and support evidence-based policy action.

### 4.5. Strengths and Limitations

A notable strength of this study is its utilization of the HLS-EU-Q16, a validated and extensively employed instrument that facilitates direct comparability with both national and international HL assessments. Its implementation in a representative multilingual sample from Alto Adige/South Tyrol provides useful insights into HL within culturally and institutionally dual regions, contexts that are often underrepresented in the existing literature. The inclusion of both index-based and domain-specific health literacy scores enables a refined understanding of the strengths and challenges across various functional areas of HL.

However, this study had some limitations. First, 21.1% of respondents had insufficient valid responses to compute a health literacy score using the HLS-EU-Q16, reducing the analytic sample and potentially introducing bias. These missing data were not missing completely at random (MCAR) but showed systematic associations with observed variables such as lower education, migration background, non-German/Italian first language, lower trust in health professionals, and lower patient activation. This pattern is consistent with a missing at random (MAR) mechanism and has been documented in prior analyses of the same dataset ([2]). To account for this subgroup analytically, cases with missing HL data were retained as a separate category in regression models, and population weighting was applied to reduce bias in descriptive estimates. Nonetheless, the possibility of residual bias due to unobserved factors cannot be excluded. Second, the cross-sectional design constrains causal interpretation. Although associations between language group, education, and HL were identified, the directionality and mediating factors underlying these relationships could not be definitively established.

Third, the regression models accounted for only a modest proportion of the variance in health literacy outcomes, indicating that essential explanatory variables, such as digital skills, health information-seeking behaviors, trust in institutions, and perceived communication quality, were not captured in the dataset. These unmeasured constructs may be particularly pertinent in multilingual or institutionally complex regions, where access and navigation are influenced not only by individual capacity but also by the responsiveness of the information environment.

Additionally, although the post hoc assignment of HLS-EU-Q16 items to conceptual domains (health care, disease prevention, and health promotion) is supported by the prior literature, this short form was not originally designed to generate subscale scores. Consequently, domain-specific analyses should be regarded as exploratory and hypothesis-generating rather than definitive.

Finally, although the study sample was demographically representative, the modest overall response rate (52.3%) may limit generalizability.

### 4.6. Future Research

Further investigation employing mixed methods, including qualitative approaches to capture the contextual influences on health literacy (HL), could enhance these findings and provide a deeper understanding of population-level disparities in health literacy competencies. In-depth interviews and focus groups can examine how individuals from diverse linguistic and educational backgrounds perceive and interact with health information, particularly concerning system navigation, trust in institutions, and digital health content. Quantitative methods could be augmented by customized survey instruments that evaluate communicative and critical HL dimensions or by linking HL data to behavioral or clinical outcomes through record linkage or longitudinal follow-ups. Additionally, participatory designs involving community co-researchers or stakeholder-driven intervention development may assist in identifying culturally grounded solutions to the HL gaps identified in this study’s findings. Such approaches are especially valuable in multilingual regions, where health communication involves not only translation but also cultural relevance, accessibility, and trust.

## 5. Conclusions

This population-based study presents the inaugural representative evaluation of HL in Alto Adige/South Tyrol, utilizing the HLS-EU-Q16, and offers critical insights into the distribution of HL across language groups within a multilingual European region. The findings indicate significant and persistent disparities in both overall and domain-specific HL, with German-speaking residents, although they form the regional majority, achieving lower HL scores than their Italian-speaking and multilingual counterparts. These language group differences remained after adjusting for key sociodemographic and health-related variables, underscoring the impact of contextual, cultural, and system-level factors on the HL.

Domain-specific analyses further demonstrated that HL is not a monolithic construct. Competencies in health care, disease prevention, and health promotion vary across groups, highlighting the necessity of multidimensional assessment and intervention. Particularly low health-promotion scores among German-speaking respondents suggest that information access, trust, and cultural alignment play crucial roles in shaping health-related competencies.

From a policy perspective, these findings challenge the assumptions of uniform health communication needs within the majority language groups and emphasize the necessity for tailored, language-sensitive strategies that reflect the sociolinguistic complexity of the region. HL must be addressed not only as an individual skill set but also as a dynamic interaction between individuals and the health information environment.

Future research should expand on these findings using mixed methods designs that integrate qualitative insights and longitudinal perspectives. Interventions aimed at improving HL should be co-developed with communities and evaluated for their domain-specific impacts across linguistic and educational groups. By doing so, Alto Adige/South Tyrol and regions facing similar challenges can progress toward a more equitable, health-literate society.

## Figures and Tables

**Table 1 ejihpe-15-00153-t001:** Sociodemographic and health characteristics according to language group (weighted proportions).

Variable	Total (*n* = 2090) ^1^	German (*n* = 1330)	Italian (*n* = 480)	Other (*n* = 280)	*p*- Value ^2^	Effect Size ^3^
Gender, % (*n*)					0.296	0.034 ^5^
Male	49.1 (1026)	49.9 (664)	49.4 (237)	44.8 (125)		
Female	50.9 (1063)	50.1 (666)	50.6 (243)	55.2 (154)		
Age (mean ± SD)	50.9 ± 17.9	50.8 ± 18.0	55.3 ± 17.9	43.3 ± 147	<0.001	0.046 ^4^
Age group, % (*n*)					<0.001	0.137 ^5^
18–34	23.7 (496)	24.0 (320)	16.7 (80)	34.2 (96)		
35–54	32.7 (683)	32.2 (429)	27.9 (134)	43.0 (120)		
55–99	43.6 (912)	43.7 (582)	55.4 (266)	22.8 (64)		
Education, % (*n*)					<0.001	0.130 ^5^
Middle school	21.1 (440)	21.5 (287)	21.6 (104)	18.1 (51)		
Vocational school	31.9 (666)	37.6 (500)	21.1 (101)	23.1 (65)		
High school	26.1 (545)	22.7 (302)	32.9 (158)	30.6 (86)		
University	20.9 (437)	18.2 (242)	24.4 (117)	28.2 (79)		
No Italian citizenship, % (*n*)	9.8 (204)	4.5 (60)	0.6 (3)	50.7 (142)	<0.001	0.544 ^5^
Work in health/social sector, % (*n*)	10.3 (214)	10.5 (140)	10.6 (51)	8.6 (24)	0.618	0.021 ^5^
Health status (mean ± SD)	76.4 ± 18.5	77.2 ± 18.4	73.3 ± 18.9	78.4 ± 18.0	<0.001	0.009 ^4^
Living alone, % (*n*)	17.8 (371)	16.7 (222)	22.8 (110)	14.2 (40)	0.003	0.075 ^5^
Chronic illness, % (*n*)	35.1 (733)	33.7 (449)	43.8 (210)	26.6 (74)	<0.001	0.111 ^5^
HLS-EU-Q16 index (mean ± SD)	11.9 ± 3.4	11.6 ± 3.5	12.5 ± 3.3	12.4 ± 2.9	<0.001	0.018 ^4^
HLS-EU-Q16 level, % (*n*) ^6^					<0.001	0.105 ^5^
Inadequate	12.6 (264)	13.9 (194)	10.8 (54)	8.3 (16)		
Problematic	27.1 (567)	28.9 (404)	22.4 (112)	26.4 (51)		
Sufficient	39.2 (820)	34.5 (482)	50.9 (254)	43.5 (84)		
Unknown/missing	21.0 (439)	22.8 (318)	15.8 (79)	21.8 (42)		

^1^ Weighted proportions and weighted counts (rounded, shown in parentheses) are reported for the total sample and each language group. Due to population weighting and variable-specific missing data, the effective sample sizes used in statistical testing may differ from the unweighted totals indicated in the table header. ^2^
*p*-values are derived from one-way ANOVA for continuous variables and chi-squared tests of independence for categorical variables. ^3^ Effect sizes are reported as η^2^ (eta-squared) for ANOVA and Cramér’s V for chi-squared tests. ^4^ η^2^ = eta-squared. ^5^ Cramér’s V. ^6^ HLS-EU-Q16 levels are based on validated index thresholds: inadequate (0–8), problematic (9–12), sufficient (13–16). The “Unknown/missing” category includes participants with more than three missing responses on the 16-item scale, rendering HL classification invalid.

**Table 2 ejihpe-15-00153-t002:** Weighted Spearman correlations between HLS-EU-Q16 health literacy index and sociodemographic or health-related characteristics in the Alto Adige/South Tyrolean population (*n* = 1651).

Variable ^1^	Spearman ρ	*p*-Value	Weighted n	Effect Size ^2^
Language: Italian (vs. German)	0.143	<0.001	1527	small
Works in health/social sector (vs. no)	0.058	0.015	1741	negligible
Education level (vs. middle school)	0.069	0.004	1741	negligible
Lives alone (vs. no)	0.049	0.043	1741	negligible
Has chronic illness (vs. no)	−0.056	0.020	1741	negligible
Gender (vs. male)	0.039	0.107	1741	negligible
Non-Italian citizenship (vs. Italian)	−0.029	0.225	1741	negligible
Age (years)	0.019	0.432	1741	negligible

Health literacy was measured using the HLS-EU-Q16 continuous index. ^1^ Reference categories are listed in parentheses. ^2^ Effect size interpretation ([8]): ρ ≈ 0.10 small; ρ ≈ 0.30 moderate; ρ ≥ 0.50 large.

**Table 3 ejihpe-15-00153-t003:** Weighted multiple linear regression predicting health literacy (HLS-EU-Q16).

Predictor	Regression Coefficient B ^1^	95% CI [Lower; Upper]	SE	t	*p*-Value
const	9.770	[8.378; 11.162]	0.709	13.781	<0.001
Language: Italian (vs. German)	0.954	[0.548; 1.360]	0.207	4.596	<0.001
Non-Italian citizenship (vs. Italian)	0.665	[−0.367; 1.697]	0.527	1.261	0.207
Has chronic illness (vs. no)	−0.506	[−0.912; −0.100]	0.207	−2.441	0.015
Works in health/social sector (vs. no)	0.494	[−0.081; 1.069]	0.293	1.688	0.092
Gender (vs. male)	0.211	[−0.149; 0.571]	0.184	1.145	0.252
Education level (vs. middle school)	0.204	[0.014; 0.394]	0.097	2.104	0.036
Lives alone (vs. no)	0.158	[−0.319; 0.635]	0.243	0.651	0.515
Age (years)	0.012	[−0.0003; 0.025]	0.006	1.905	0.057

The outcome variable was the continuous HLS-EU-Q16 index (range: 0–16). Regression coefficients (ß) were estimated using weighted least squares (WLS) regression. CI, confidence interval; SE, standard error. ^1^ Unstandardized coefficient (B) is reported, representing the expected change in the outcome variable per unit change in the predictor variable.

**Table 4 ejihpe-15-00153-t004:** Weighted distribution of domain-specific HLS-EU-Q16 scores by language group in the Alto Adige/South Tyrolean population (*n* = 1649).

Domain ^1^	Language Group	% Top Score ^2^	χ^2^ (df) ^3^	*p*-Value	Cramér’s V ^4^
Health care (top score = 5)	German	26.3	49.24 (20)	<0.001	0.084
	Italian	40.2			
	Ladin	37.0			
	Other	35.4			
	More than one language	28.3			
Disease prevention (top score = 4)	German	47.2	51.08 (16)	<0.001	0.086
	Italian	62.4			
	Ladin	61.6			
	Other	60.4			
	More than one language	69.6			
Health promotion (top score = 7)	German	23.7	45.57 (28)	0.019	0.081
	Italian	33.6			
	Ladin	21.9			
	Other	36.5			
	More than one language	37.0			

^1^ HL domains were derived from the HLS-EU-Q16 by grouping items into the theoretical domains of health care (five items), disease Prevention (four items), and health promotion (seven items), as described in the Methods section. ^2^ Percentage of respondents within each language group who achieved the maximum score in that domain. ^3^ Chi-square tests of full ordinal score distributions across language groups; ^4^ Cramér’s V as a measure of effect size.

## Data Availability

The original data presented in this study are available from the corresponding author upon request.

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
