# Peer review of "Health Literacy Gaps Across Language Groups: A Population-Based Assessment in Alto Adige/South Tyrol, Italy"

_ejihpe, 2025, doi:10.3390/ejihpe15080153_

Round 1
Reviewer 1 Report
Comments and Suggestions for Authors
This is a clearly structured and well-written manuscript describing a cross-sectional study aiming at exploring health literacy (HL) differences between language groups in the region of Southern Tyrol. I have only minor comments.
The HLS-EU survey instruments are not according to my evaluation truly validated questionnaires but more based on carefully pre-planned developmental work. True validation studies would according to my opinion imply a study where the survey data would be compared with corresponding data collected with another, as reliable as possible, methodology.
In row 114 ‘…sociodemographic predictors..’, in a cross-sectional study I would prefer the term ‘determinants’ instead. The same applies, naturally, to row 206 and elsewhere in the manuscript.
It remains somewhat unclear in which analyses the respondents that responded to 13 – 16 items of the HL items were included although it is stated that they were excluded from the correlation and regression analyses but if they were included in some analyses, how were the subcategory sum scales calculated?
In the primary results it can be seen that HL was quite high in the language ‘More than one’ group. Naturally, most probably it is a very heterogenous group but I would have been interested in further analyses based on the hypothesis that individual multiculturalism might promote HL. However, I can see that this would imply another study with different aims as compared to the present study.
In row 112 ‘…LH levels in South Tyrol…’ should be HL.
Reviewer 2 Report
Comments and Suggestions for Authors
Dear Authors,
Congratulations on completing the manuscript and conducting such an interesting study. However, the manuscript still needs some improvements. I have the following recommendations and suggestions to increase the value of your study.
The manuscript as a whole suffers from an inconsistent structure and ambiguously defined study aims. The information in lines 76–85 suggests that the reason for the below-average values in the Italian sample is the phrasing of certain items in the language version of the questionnaire or cultural and cognitive response patterns. If so, a study based on cognitive interviews comparing the two language versions of the questionnaire should have been conducted. However, representative research was conducted in a specific location (South Tyrol), which seems inappropriate from a methodological point of view. Therefore, I request a careful and comprehensible explanation of the study's purpose and the reason why representative research was conducted on the given population. Additionally, several assumptions have not been adequately tested or verified in the existing research (see, for example, the speculation in lines 94–95). Based on information provided elsewhere in the introductory text, it appears that the study aimed to conduct descriptive research in a specific location — see, for example, the statement on lines 101–102. Another issue with the research design is the mention in lines 104–105 of another research design. However, this design was clearly not used; therefore, the objective could not be achieved using the existing results. Due to the unclear objective, it is impossible to unequivocally assess whether the chosen methods are adequate.
Introduction:
Lines 62 and 74 refer to HLS19, but the data relating to Germany are based on HLS-GER 1 (line 68), yet the relationship between the cited data sets is not properly explained. Please add an adequate explanation.
Paragraphs 106–116 would be more appropriate in the conclusions section. Please revise the content or move it.
The introduction lacks a clear formulation of the research question and the hypotheses posed by the authors. Please add them.
Methods:
Based on the information in paragraphs 131-135, the weighting method must be described, including the criteria used to weight the data, the method used, and the range of weight values. Ideally, include the structure of the sample before weighting, the structure of the weighted sample, and the structure of the theoretical population in the appendices. Table S3 raises significant doubts about the correct weighing method, as it indicates an unusual increase in n to 2,219, though the actual sample size was 2,090 respondents, according to line 218.
2.5 Statistical Analysis
Due to the unclear objective and purpose of the study, it is difficult to assess the appropriateness of the analytical methods used. Once the objective is clarified, a subsequent review of the data analysis methods should be conducted (are inferential statistics really necessary?).
Regarding the results the study aims to achieve, I find the application of SEM lacking (see lines 385–386, for example). Since the research task clearly calls for SEM, I request an additional SEM analysis and adequate description of this method in section 2.5.
Results
Please provide an additional explanation of the difference in composition between the Italian population and the other subsets. The data in Table 1 show that the Italian population is, on average, 4–12 years older. There is also a higher prevalence of chronic illness and single-person households. These facts may be significantly related to health literacy.
It is certainly not possible to conclude that a Cronbach's alpha value of 0.566 is acceptable, and the vast majority of relevant sources confirm this. It is definitely not! The same applies to values lower than 0.7.
Due to the different bases of the individual analyses (2,090 in line 218, 1,651 in Table 2, and 1,436 in Table 3), please comment on the nature of the missing cases (MCAR, MAR, or MNAR). I would like to point out that the explanation in lines 556–563 is not specific enough.
Due to their interpretative nature, paragraphs 373–379 belong more in the discussion than in the section presenting the results.
Discussion:
Please explain why the "linguistic appropriateness of health information" was not measured when it was predicted (see lines 82–83).
Indicate whether the finding in lines 427–428 is generally valid or applies only to South Tyrol.
However, the information on the "similar demographic profile" mentioned in line 448 contradicts the data in Table 1. The other information in this paragraph (lines 447–455) is mere speculation not supported by the presented data. This irrelevant information misses the purpose of the "Discussion"; therefore, please thoroughly revise the text.
Explain how the information in lines 501–505 relates to the objectives of the study indicated in the introduction.
Regarding lines 511–520, do you have any empirical evidence that this is not the case?
I have pointed out some key issues based on a careful analysis of your paper. Please consider carefully revising your manuscript to increase its chances of positive acceptance by the professional community. I wish you all the best in your future work.
Sincerely yours,
Round 2
Reviewer 2 Report
Comments and Suggestions for Authors
Dear Authors,
I would like to express my appreciation for the efforts you have made to revise the manuscript. It is clear from your revisions that you have thoughtfully considered all of the comments, and the manner in which you have addressed these issues is commendable. The changes you have made demonstrate an effective approach to improving the quality of your work. After reviewing the updated manuscript, I am pleased to write that the revisions meet the expectations set forth in my previous comments. The changes you have made have improved the clarity and overall impact of the study. The manuscript has been refined to a standard that effectively communicates the research findings and provides valuable insights to the field.
Sincerely,